# Piloting an Information and Communication Technology Tool to Help Addressing the Challenge of Antimicrobial Resistance in Low-Income Countries

**DOI:** 10.3390/antibiotics14040373

**Published:** 2025-04-03

**Authors:** Florence Mutua, Joshua Orungo Onono, Sofia Boqvist, Patricia Koech, Abdullahi M. Abdi, Hildah Karimi, Susanna Sternberg-Lewerin

**Affiliations:** 1International Livestock Research Institute, Nairobi 00100, Kenya; f.mutua@cgiar.org (F.M.); patcherotich22@gmail.com (P.K.); abdullahimohaa4459@gmail.com (A.M.A.); kirimihildah3@gmail.com (H.K.); 2Department of Public Health, Pharmacology & Toxicology, University of Nairobi, Nairobi 00100, Kenya; joshua.orungo@uonbi.ac.ke; 3Department of Animal Biosciences, Swedish University of Agricultural Sciences, 75007 Uppsala, Sweden; sofia.boqvist@slu.se

**Keywords:** antimicrobial use, smallholder, poultry production, veterinary, pharmaceuticals

## Abstract

**Background/Objectives**: Antimicrobial use (AMU) in livestock drives antimicrobial resistance (AMR). AMR has a significant impact on public health. While several interventions have been used to address this challenge, few have utilized Information and Communication Technology (ICT) approaches. The objective of this study was to pilot and assess an ICT system to monitor the use of veterinary drugs and disseminate information to farmers in peri-urban smallholder poultry systems in Kenya. **Methods**: The system was developed in collaboration with the stakeholders. It captures drug sales in veterinary pharmacies and disease incidence and treatments reported by farmers. The system was piloted from May 2023 to December 2023. Monthly follow-ups were conducted to monitor progress and address problems. Assessment was performed through focus group discussions with the users (two with farmers and two with veterinary pharmacy staff) and descriptive statistics of the data collected by the system. **Results**: A total of 15,725 records were obtained from veterinary pharmacies, including antibiotics (57%), dewormers (22%), and vitamins (11%). Requests for a specific product were recorded in 38% of the sales, while 63% were accompanied by some evidence (empty drug containers, old packages, old prescriptions, pictures of sick birds, and actual sick birds). A total of 91 records were obtained from the farmers. The health problems reported were mostly respiratory (40%) and digestive (30%) disorders. The percentage of customers who requested advice on animal health when visiting veterinary pharmacies ranged from 5 to 20%. **Conclusions**: AMU can be improved in the study area. The piloted system may help policymakers monitor the sales and usage of antibiotics, improve animal health management, and promote responsible AMU.

## 1. Introduction

Antimicrobial resistance (AMR) causes more than 700,000 deaths annually [1], and the burden is particularly high in the WHO African region [2]. In the presence of antibiotics, bacteria respond to this selection pressure by acquiring and/or expressing antibiotic resistance genes (ARGs), and antibiotic use drives the dissemination of mobile ARGs among bacterial populations [3]. Bacteria and their genes are easily transmitted within and between human and animal populations and the environment [3]. Antibiotics are widely used to treat bacterial infections in animals intended for food production [4]. They are also used for general animal health promotion and to increase livestock production [5]. In East Africa, farmers frequently fail to observe withdrawal periods, which results in the presence of antibiotic residues in animal products [6]. Such residues may cause allergies, toxic reactions, and promote AMR [7]. In addition, antimicrobial use (AMU) in livestock production contributes to AMR in humans [8]. Antimicrobials should be used sparingly in humans and animals to avoid unnecessary selection pressure for AMR [9]. In this respect, the use of antimicrobials as growth promoters is of particular concern [10].

The need to strengthen the knowledge and evidence base through surveillance and research was one of the objectives outlined by the World Health Assembly in 2015 to combat AMR globally [11]. Surveillance data on AMR and AMU in both animals and humans are critical for decision-making, identifying areas that should be prioritized, and following up on interventions. There have been calls to institute surveillance measures to understand the extent of AMU in the agricultural sector [9]. Although there has been progress [12], available evidence shows that most countries in Africa do not have national AMU and AMR surveillance systems and rely on point-prevalence studies [13]. The lack of surveillance data makes it difficult to comprehend the true scope and impact of AMR [14]. AMR surveillance in Kenya’s animal health sector began in May 2021, with six laboratories reporting data to a common national database [15]. There is currently no national surveillance system for AMU or consumption in the Kenyan human health sector [15], and the same applies to the animal health sector. Veterinary pharmaceuticals are registered in the Kenyan government system for imports and exports; however, there is no system for collecting AMU data at the points of sale or at the farm level.

Evidence from recent studies in Kenya shows that veterinary drugs are often administered by the farmers themselves, and often without veterinary prescriptions [16,17,18,19]. This practice increases the risk of AMR, and there is a need to initiate surveillance of AMU in Kenyan livestock production systems and improve awareness of AMR among livestock owners. Information and Communication Technology (ICT) could be a useful tool in Kenya, given the high use of smartphones, good coverage of mobile networks, and access to solar-powered charging equipment. A system for veterinary reporting of notifiable animal diseases via smartphones has already been developed and successfully implemented in most parts of Kenya [20]. Against this background, we developed an ICT application (ADIS), animal disease information system, to monitor the sales of veterinary drugs at the level of veterinary pharmacy outlets (agrovet shops) and to disseminate information to poultry farmers [21]. The development and piloting of the system have been described [21]. The current study aimed to further assess its usefulness and provide evidence of AMU practices that national and local policymakers can use to develop strategies for AMU monitoring and AMR control in the country.

## 2. Results

### 2.1. Farmer Pilot Preparation Survey

Only 15 farmers were available and interviewed in the preparation survey, which was conducted before launching the pilot study. The majority were female (11/15) and kept improved breeds of poultry (45%; n = 24, considering one could keep multiple types). An improved breed is a cross between a local/indigenous bird and an exotic breed. The health problems reported included one case of chicken pox, five cases of sudden death, three cases of infectious coryza, two cases of fowl pox, two cases of dropping feathers, one case of eye infection, one case of swollen leg, and two cases of flea infestation. A health professional was consulted in the case of chicken pox, diarrhea, infectious coryza, fleas, dropping feathers, and sudden death. Two farmers used medicines to prevent infections in the preceding month.

A total of 60 treatments were described by the 15 farmers; 16 of these were vitamins (27%), 15 antibiotics (25%), 8 dewormers (13%), 7 acaricides (12%), 5 disinfectants (8%), 8 other (not specified) (13%), and one that was stated as unknown. Aliseryl, Amprolium, Esb, Fosbac, Heltivet, Oxysol Plus, Tetracolivit, Tylodoxy, and Veta-Oxy were the antibiotics mentioned by the farmers. Almost all (96%) said that the products they had used worked, a majority of which (96%) were administered by the farmers themselves, mostly through drinking water (76%). Twelve treatment records (20%) were accompanied by veterinary prescriptions (three of which were reported as antibiotics). Leftover drugs were described for 16 treatments (27%), and in one of these, the drug was reportedly disposed of. Packages and empty containers were mostly (71%) disposed of by burning. Other methods included burying, throwing in pit latrines, cleaning and re-using containers. In 47 treatment events (78%), the eggs and meat produced in the two weeks following the treatment period were sold or consumed, while in four (6%) cases, the products were fed to animals.

### 2.2. Farmer Data Recorded in the ADIS System

Ninety-one (91) records were received from the farmers, 12 records had missing details in the variable health problems (either only stating ‘new symptom’ or the name of a drug) and were excluded from the analyses including this variable, reducing the total number of analyzed records to 79. Most reported health problems were respiratory (40%, n = 32) and digestive (30%, n = 24) (Figure 1). The digestive problems were mainly described as bloody diarrhea (75%, n = 18).

Forty-eight treatment records were found: 26 antibiotics (54%), 17 dewormers (35%), and 11 vitamins (22%). Table 1 lists the products used to treat various health conditions.

### 2.3. Agrovet Data Recorded in the ADIS System

A total of 15,725 records were entered by 14 participating agrovets. The system required entering information about the location where the purchased drugs would be used. However, 4% (n = 629) of the records did not contain exact information about where the products were used. The majority were to be used within the two counties (Figure 2). However, some drugs (950 records) were also sold for use in counties outside of the study area (Figure 3).

The reasons why farmers visited the agrovets included having sick animals (58%), wanting advice on disease control (20%), wanting advice on animal husbandry (11%), and purchasing products for disease prevention (10%). In 38% of the 15,725 recorded sales, customers requested a specific product. The drugs were intended for use in different species of livestock (Table 2), including camels, cats, cattle, dogs, aquaculture, goats, and poultry.

About 63% (n = 10,013) of the drug purchases were accompanied by some form of reference. Figure 4 shows the references used to support drug purchases for poultry treatment. In 35 records, the farmer showed a sick bird or a picture of a sick bird, but the product sold was for other species. New prescriptions were available in 65% of the total (n = 6884) of the prescription records.

Over half of the 15,725 records (57%) were antibiotics (interpreted as any drug found to have an antibiotic in its composition). The drugs commonly sold are summarized in Figure 5. Aliseryl^®^, Tylodoxy 200^®^, and Trimovet^®^ were among the most frequently sold antibiotics (Figure 5). The dewormers (22%) included albendazole, fenbendazole, ivermectin, levamisole, praziquantel, nitronil, piperazine, mebendazole, and ricombendazole. Vitamins constituted 11% of the records, while mineral supplements constituted 1.4% of the records. Antiprotozoal products (5%) included amprolium, buparvaquone, parvaquone, diminazine aceturate, imidocarb dipropionate, isometamidium etc.

### 2.4. Perceptions and Implementation of ADIS by the Agrovet Attendants

The agrovet attendants said that ADIS encouraged them to keep records of what they sold. The ADIS system allowed them to obtain an overview of the drugs used in their localities. In the study, they were able to follow up with the customers and get to know the outcome of using the prescribed drugs (i.e., if the treated animal recovered).

The percentage of agrovet customers requesting advice on animal health was said to range from 5 to 20%. A few attendants said they were able to view the disease information that was part of the farmer module. The system enabled them to obtain the correct disease history from the farmers (which, they said, would at times lead to a reduction in the use of antibiotics and consequently decrease sales). The main perceptions that emerged from the FGDs are presented in Table 3.

The percentage of sales captured by the system ranged from 75 to 100%. Busy times included the end of the month, holidays, and sometimes weekends. Market days were said to have more sales than non-market days. The months of November and December (2023) had low sales, reportedly due to a lack of day-old chicks in the market. The recording in ADIS, if done at the time of purchase or later, depended on the day’s schedule. When busy, the attendants would record the sales in a book and transfer them to the system later. When not busy, they entered the data immediately. Although one person in each agrovet shop was responsible for entering the data, their colleagues assisted when they were away from the shop. Some said they would forget to record some sales. The attendants said they would be interested in using the system in the future, not only for record-keeping but also to trace their customers and track drug sales.

According to the agrovet operators, the main change that occurred since the start of the pilot study was the introduction of new pharmaceutical products in the market.

### 2.5. Perceptions of ADIS and Treatment Practices of the Poultry Farmers

The farmers said the ADIS App enabled them to interact among themselves (perhaps in the meetings, since the app itself did not support interaction between farmers). They reported that they were able to consult animal health providers through the app. The farmers also stated that the information provided was educative, and they did not have to rely on the agrovet operators. The farmers reported that they were able to report the disease issues on their farms in ADIS, as well as the drugs they used to treat the problems.

On the question of how long the farmers wait before deciding to treat sick birds, three categories emerged: those that immediately treat the birds, those that will wait for about three days, and those who do not treat until they have observed mortality among the birds. The farmers were further asked to say how they could tell when the given medicine was working (or not working), and varied responses were received, such as when the disease symptoms have subsided, when the birds become alert and energetic (exhibit normal/natural behavior), when the birds start feeding, or after a reduction in mortality. The second part of this question sought to understand how long farmers usually wait before concluding that the treatment has not worked for them. Some participants would wait for 2–3 days, but some would observe the drug dosage time and make the decision once this time had passed. Their thoughts on how unnecessary use of drugs can be reduced included: hygiene in the farms; measures to ensure that biosecurity is maintained in the farm; herbal formulations from trees such as aloe vera, ‘mukinduri’, ‘mwarubaini’, and from pepper and charcoal; and adhering to vaccination schedules.

The major changes that occurred on the farms after the project began included a reduction in flock size attributed to a scarcity of day-old chicks in the market, changes in the breed of chicken on the farms (from layers/broilers to indigenous or improved ‘kienyeji’ and vice versa), and some farmers who reportedly lost their poultry to disease outbreaks.

## 3. Discussion

Our study included farmers and agrovet operators, both of which are critical for disease control and AMR prevention strategies. While the number of participants was low, it was considered appropriate for the study because regular follow-up visits to provide technical support were foreseen. Farmers used the ADIS to report symptoms they encountered on their farms. The observed symptoms could be indicative of important diseases and, along with laboratory testing, could guide interventions on the farm and in the area. For example, bloody diarrhea can be due to coccidiosis, which has negative impacts on farm productivity. For general disease surveillance, syndromic surveillance is easier to implement, especially in resource-poor settings, as it relies on the identification and quantification of specific symptoms as signs of a possible outbreak, with no strict requirements for a specific diagnosis [22]. This form of surveillance utilizes data that precede diagnoses and provides signals of a potential case or outbreak that warrants further public health responses [22]. Nevertheless, syndromic surveillance relies heavily on the willingness and ability of farmers to report symptoms. This, in turn, assumes that farmers are able to recognize and correctly report relevant symptoms at an early stage. In addition, fear of negative consequences, such as government restrictions on farms, may prohibit farmers from reporting.

Our study established different reasons why farmers visit agrovet outlets, including when animals are sick and to obtain advice from the attendants on husbandry and disease control. While this highlights the central role that agrovets play in disease control, there are some concerns. Farmers often visit agrovet shops without veterinary prescriptions, describe their animals’ symptoms, and are sold drugs based on this interaction with the staff. However, they can also request a certain product that they have used before (or that has been recommended by their neighbors). Given that the animal has not been examined by a veterinarian and the diagnosis is not confirmed, this behavior can promote unnecessary use of antibiotics. In current Kenyan legislation, category 2 veterinary medicines (including most antibiotics) require a prescription from a registered veterinary practitioner [23]. Veterinarians play a vital role in managing AMR because they frequently prescribe antimicrobials to protect animal health [24].

Farmers bringing sick birds to agrovet shops is dangerous as it can cause disease spread. Increasing veterinary consultations by farmers and training in sample collection and transfer to laboratory facilities could reduce the risk of disease outbreaks.

Poultry farmers often administer drugs themselves [18], as confirmed in this study. While certain factors may push farmers to use antimicrobials, unnecessary use will contribute to AMR without benefits and with negative implications on health. Resistant bacteria can spread through poor hygiene practices in healthcare and agriculture, as well as through unsanitary conditions in the food chain. Antibiotics are vital for animal health and cannot be replaced in the near future [4]. AMR is a particular challenge in Africa, where access to appropriate therapy is limited, regulations governing the use of antimicrobials for humans and animals are weak, surveillance systems are lacking, and guidelines for antimicrobial use and treatment are lacking [13,25].

The pilot study involving agrovet outlets yielded over 15,000 records. The data show that the majority of the sold drugs were going to be used within the study counties, which is not surprising as these were the main sites for the study. However, about 6% of the included drugs were used in other counties. Even in the study counties, the drugs were used in places outside of the areas where the study was implemented (i.e., Machakos Central/town and Kajiado North). This information is useful for surveillance and informing future interventions. The agrovet platform captured data on the quantity of drugs sold during the study months. The trend indicates that there were months when more drugs were sold. This could serve as an early warning sign for important diseases and alert authorities to respond and investigate what might be prompting the increased use (i.e., if there is a disease outbreak). Fosbac^®^, Tylodox^®^, Limoxil^®^, Tylodoxine^®^ and Tylosine 75^®^, which were among the most frequently reported drugs in the current study, have been described as “magic” [26] and are thus more likely to be misused by non-professionals. Thus, there is a need to create awareness, change perceptions, and highlight the need to engage qualified professionals. Regardless of their professional responsibilities, reduced sales of antimicrobials could mean reduced income for agrovet operators; hence, an incentive to promote the use of a reporting system is needed. This could be in the form of subsidies for the implementation of advisory services that reduce antimicrobial sales and promote alternative sales, such as products for cleaning and disinfection, vaccination, and good animal husbandry.

Our study piloted an ICT intervention to monitor the use of veterinary drugs in peri-urban smallholder poultry systems. A previously introduced smartphone-based surveillance system for notifiable animal diseases has been successfully implemented by government authorities in Kenya [20]. Experiences from that implementation and our own work [21] demonstrate the need for offline data entry during poor internet connectivity. The data provided in this study demonstrate the potential of the ADIS to monitor AMU and contribute to AMR risk reduction. However, further work is needed to determine how the system can support ongoing surveillance work as a whole, or specific elements that stakeholders prioritize. If the national, regional, and local authorities assume ownership of the system, it could be further developed and implemented based on regulatory requirements. Before mandatory implementation, extensive training of the intended users, as well as a thorough assessment of potential incentives and disincentives and how to address these, should be performed.

The drug purchase and use practices observed in this study present a risk for AMR and require urgent attention. In Kenya, the law that regulates the operations of veterinary pharmacies is the Veterinary Surgeons and Veterinary Paraprofessionals Act [23]. In this legislation, the requirements for veterinary prescriptions were strengthened. The responsible authorities are currently working on improving compliance with this legislation through nationwide inspections and knowledge dissemination activities, some of which have been supported by our project [21]. Addressing the risk of AMR requires a One Health approach; hence the need to engage and collaborate with all relevant stakeholders. A key consideration for future improvement of the tool is to develop an approach to validate the quality of data submitted through ADIS.

Although Kenya has established a good foundation for addressing AMR, more support is needed to realize its impact [15]. Tools such as those piloted in our study are timely for addressing this challenge.

## 4. Materials and Methods

### 4.1. Study Sites

The study was part of the JPIAMR project “MAD-tech-AMR- Management of animal diseases and antimicrobial use by information and communication technology to control AMR in East Africa” (https://www.jpiamr.eu/projects/mad-tech-amr/, URL accessed on 11 March 2025). The study was implemented in Machakos and Kajiado counties of Kenya, which were chosen because of their convenient location and good collaboration with the local veterinary authorities. A detailed description of the study areas can be found in Mutua et al. [18]. The main towns at the two sites are approximately 60 km from Kenya’s Capital City, Nairobi. The study involved veterinary pharmacies (hereafter called agrovets) and poultry farmers. Agrovets are shops authorized by the Veterinary Medicines Directorate (VMD) to sell agricultural products, including veterinary drugs, under the management of qualified veterinary professionals. The VMD is the agency mandated to regulate the sale of veterinary medicines in Kenya [23].

### 4.2. Selection of Study Participants

This study was a follow-up to the baseline activity implemented in 2022 [18]. Farmers included in the current study were selected from the 100 who participated in the baseline survey based on owning a smartphone, having access to the Internet, and willingness to participate in the follow-up activity. A pilot preparation survey with a face-to-face administered questionnaire was conducted with the selected farmers and was designed to capture data on the health problems encountered on the farms, drugs used to treat the birds, and the methods used to dispose waste resulting from the use of the drugs (packages, etc.). Agrovet operators were also identified from the list engaged in the baseline study, considering their availability and willingness to participate in the follow-up study.

### 4.3. Piloting the ICT System

The development of the ICT system, challenges encountered, and general user perceptions during the piloting of the system have been described previously [21]. It was named the Animal Disease Information System (ADIS). Briefly, it consists of a platform for agrovets to register drug sales (agrovet module) and a mobile application for farmers to report animal disease and veterinary drug use (farmer module), linked to a database to monitor drug sales and provide general information about animal husbandry, animal health, and AMR.

The study was conducted between May 2023 and December 2023 with support from County Government officials. Agrovet operators were contacted and informed about the pilot activity (which they already knew about from previous meetings). Consent to participate was sought when the research team visited their places of work. Those who agreed to participate were provided with tablets and were subsequently enrolled in the study. A demonstration of how to interact with the ADIS was provided, mostly to the person who was going to be actively involved in capturing the data. The data entries required for each sale are listed in Appendix A, Table A1. Any questions were addressed by the project team. A date to start data collection was agreed upon during this first meeting (the majority had started within a week of enrollment).

A visit to each farm was arranged, during which a mobile application (app) developed for ADIS was installed on their smartphones. A demonstration of how to use the system was subsequently provided, and any questions were addressed by the project team.

The piloting work was conducted for a period of 6–8 months. Bi-weekly field visits were arranged for both the farmers and the agrovet outlets (these were later reduced to once every month). The visits were meant to monitor progress and respond to concerns raised by the participants or issues that the research team had observed from the database. Participants were encouraged to contact the research team if they encountered any challenges while interacting with the ADIS.

### 4.4. Data Collection

As we aimed to demonstrate that ADIS could not only be used to capture data on AMU but also disseminate information on biosecurity and other disease control practices, a final data collection exercise was conducted in February 2024, which included FGD meetings with participating farmers and agrovets attendants. Separate checklists were used for each target group (see Appendix B). Two meetings were arranged in each county, one with farmers and another with agrovet operators. The discussions were led by a moderator (female with postgraduate training) and an assistant who took notes and recorded the discussion. Interviews were also conducted with key informants (i.e., government and county officials and veterinarians conversant with the project) to learn about their perceptions of the system and how it could be improved. The questions were tailored to suit each key informant. As for other data collection activities, the study was explained and consent was sought before the discussions began. The findings were discussed and validated in a meeting with stakeholders in April 2024.

Data from the ADIS records were downloaded as MS Excel^®^ (Microsoft Co., Redmont, Washington, DC, USA) files. All data were checked for consistency and cleaned before analysis. The total pack size of the drug was computed as the package size (e.g., 100 g) multiplied by the quantity bought (e.g., one package). Missing entries included those in which the details on quantity were not clear. The analyses were mostly descriptive (tables, graphs).

## 5. Conclusions

Our data demonstrate that there is room for improvement in AMU in the study area. The piloted system may help policymakers monitor sales and usage of antibiotics, improve animal health management and promote responsible AMU. Further work is needed to determine how the system can support ongoing disease surveillance work and what specific elements stakeholders prioritize. In addition, methods for data quality assessment should also be explored.

## Figures and Tables

**Figure 1 antibiotics-14-00373-f001:**
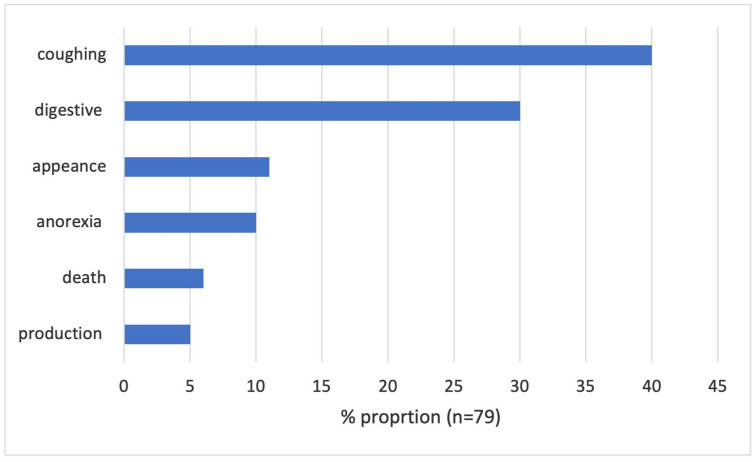
Health problems reported by farmers in the ADIS system (May–December 2024).

**Figure 2 antibiotics-14-00373-f002:**
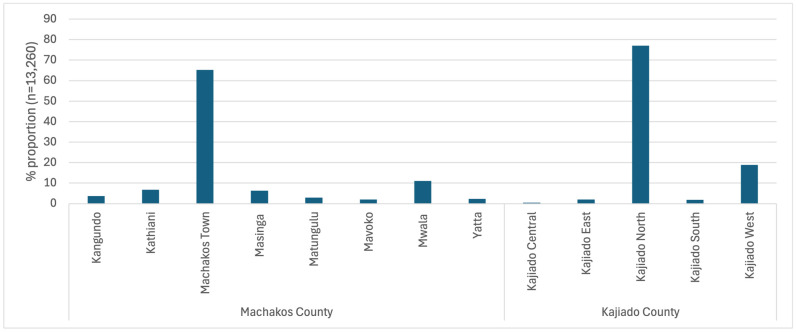
Sub-counties where the drugs purchased from the study agrovets (from within each county) were reportedly used (n = 13,260) (May–December 2024).

**Figure 3 antibiotics-14-00373-f003:**
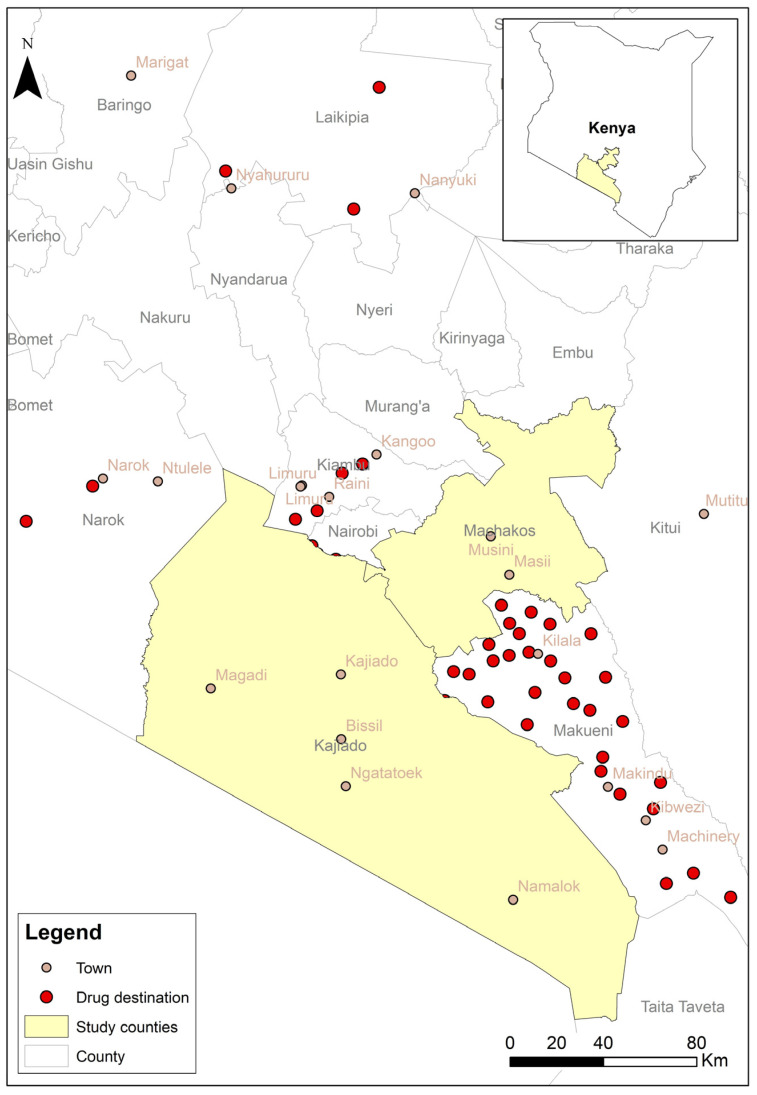
Areas outside their county, where drugs sold by agrovets in Machakos and Kajiado Counties were reportedly sold (May–December 2024).

**Figure 4 antibiotics-14-00373-f004:**
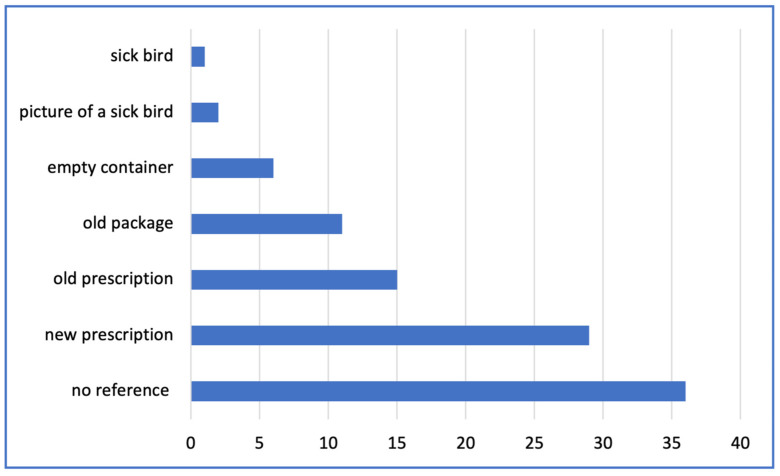
References provided to support the purchase of pharmaceutical products for poultry (May–December 2024).

**Figure 5 antibiotics-14-00373-f005:**
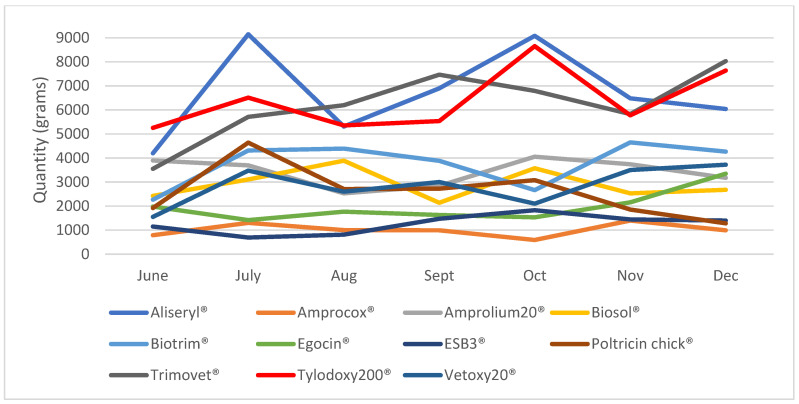
Estimation of the quantity of drugs sold from June to December 2023. Product content: Aliseryl (erythromycin, oxytetracycline, streptomycin, colistin sulfate); Amprocox (Amprolium, Sulfaquinoxaline); Amprolium (amprolium); Biosol (trimethoprim, Sulphamethoxazole); Biotrim (Sulphamethoxazole, trimethoprim); Egocin (oxytetracycline); ESB3 (sulfaclozine); Poltricin chick (Vitamins A, D3, E, B12, riboflavin, niacinamide, pantothenic acid); Trimovet (sulphamethoxazole, trimethoprim); Tylodoxy 200 (tylosin tartrate, doxycycline); Vetoxy 20 (oxytetracycline).

**Table 1 antibiotics-14-00373-t001:** Drug use practices as reported in the ADIS system by the poultry farmers (May–December 2024).

Drugs Reported in the System	Drug Category	Treatment Cause
Tylonor 20, KX-Doxytylosin, Gentamox, Ashoxy chick formula ^1^, Bilosin, Betamox, Collie-AM, Supermed TS ^2^, Pen & Strep, Tetracycline 25%, Ashtyl, Colivet-4800, Vapcotrim powder, Ashoxy egg formula	Antibiotics	data
New-Pacprim	Vitamins	Coughing, diarrhea
Alben & Iver Oral, Tectin, Bimectin	Antiparasitics	Lice, wet feces, not eating, worms in feces, gas, something new

^1^ Also contains vitamins; ^2^ contains macrolides, beta lactams, aminoglycosides, sulfonamides, and tetracycline.

**Table 2 antibiotics-14-00373-t002:** Livestock species for which the drugs sold by agrovets enrolled in the study were to be used (May–December 2024).

Animal Species.	Sold by Agrovets in Machakos (n = 4643)	Sold by Agrovets in Kajiado (n = 7832)
Camel	1 (<1%)	2 (<1%)
Cat	25 (<1%)	48 (<1%)
Cattle	1010 (22%)	2267 (29%)
Dog	263 (6%)	470 (6%)
Fish/aquaculture	3 (<1%)	15 (<1%)
Goat	283 (6%)	780 (10%)
Poultry	2962 (64%)	4111 (52%)
Rabbit	96 (2%)	139 (2%)

**Table 3 antibiotics-14-00373-t003:** Perceptions of agrovet attendants emerging in FGDs after pilot testing of an ICT system in Kenya, May–December 2024.

Key Aspect	Main Points
Capacity of agrovet attendants to advise customers on animal health	Attendants are well able to advise their customers, but may sometimes need to consult a colleague (veterinarian or paraveterinarian). Attendants are able to maintain the balance between professionalism and business improvement (by prescribing alternatives to antibiotics). They incur losses if they do not sell drugs. Attendants have a responsibility to advise customers on the right dosage but face challenges when farmers present no prescription.
Response to customers requesting specific medical products	Attendants reportedly ask for the case history and redirect them to the appropriate drug. Some farmers refuse to take the advice and are then sold what they have requested.
Thoughts on how AMU can be optimized and overuse/misuse reduced	Regulators should enforce implementation of the regulations. Agrovet attendants should be professional and sensitize the public on AMR, including use of alternatives. Ways of tracing animal products to their production source should be explored. Advertising for antibiotics/antimicrobials on social media should be banned. Infographics could be presented in posters for the farmers to read when they visit the shops.

## Data Availability

The data generated during the current study are not publicly available due to a promise of anonymity to the study participants, but can be obtained from the first author in an anonymized format upon reasonable request.

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
