# Peer review of "Piloting an Information and Communication Technology Tool to Help Addressing the Challenge of Antimicrobial Resistance in Low-Income Countries"

_antibiotics, 2025, doi:10.3390/antibiotics14040373_

Round 1

Reviewer 1 Report

Comments and Suggestions for Authors

The abstract provides a clear and concise summary of the study, effectively outlining its objectives, methods, results, and conclusions. However, it would be better if additional details on how the system was evaluated, such as the number of qualitative interviews conducted are included in the methodology sub-section.

While the authors have provided a strong justification for AMR surveillance in introduction section, it would better to articulate the specific gaps in Kenya’s current AMU monitoring efforts, particularly in livestock farming, to emphasize the novelty and necessity of the study. Additionally, while the authors have briefly mentioned the risk of antimicrobial residues in animal products. The potential health consequences for consumers have not been discussed in enough detail. Moreover, while the role of ICT in AMU monitoring is introduced, there is limited discussion on previous digital interventions in similar settings or their potential limitations. Additionally, the transition from the problem statement towards the study’s objectives could be smoother to ensure a much natural flow from the broader AMR issue towards the specific rationale of piloting ADIS.

In the results section, while the authors have presented a significant amount of numerical data but there are some inconsistencies in how certain values are reported (such as switching between percentages and absolute numbers), which could make interpretation challenging for readers. The sub-headings particularly those which summarize the farmer and agrovet perceptions, are quite descriptive but could benefit from a more structured analysis that groups findings into broader themes making it easier to identify key takeaways.

In the discussion section, while the authors have highlighted the critical role of farmers and agrovet operators in disease control and AMR prevention but they have not fully discussed the challenges of implementing syndromic surveillance such as potential misinterpretation of symptoms or reluctance among farmers to report cases accurately. The authors have also mentioned the economic implications for agrovet operators due to reduced antimicrobial sales but could have further discussed the potential policy or financial solutions to balance public health concerns with their livelihoods. The discussion section could also be more beneficial if a clearer linkage between the study’s findings and existing AMR policies in Kenya, as well as practical steps for integrating ADIS into national surveillance frameworks are discussed.

The methodology section is well-structured. However, while the authors have described the study sites. But there is little discussion on why these specific counties were chosen beyond their proximity to Nairobi. The piloting of ADIS is also well explained but it is unclear whether any technical challenges were encountered such as issues with user compliance, data entry errors or system reliability. Additionally, the data collection process is mentioned but authors should also mention regarding any validation methods if applied to ensure accuracy in self-reported drug use and disease cases.

In the conclusion section, the authors could also briefly mention the recommendations for future research or steps needed for scaling up the system such as refining the data collection process, increasing farmer participation, or integrating the system with existing surveillance programs.

I have also been noted that there are some grammatical issues in the manuscript. For example, authors should consider the subject-verb agreement error "Evidence from recent studies in Kenya show" and replace it with "Evidence from recent studies in Kenya shows". In the conclusion section "improvement of AMU" should be revised to "improvement in AMU". Similarly, "help policymakers to monitor" should be changed to "help policymakers monitor". Additionally, the phrase "and responsible AMU" is vague and would be clearer if it is modified as "and promote responsible AMU."

Comments on the Quality of English Language

The English could be improved to more clearly express the research.

Author Response

Comment 1: The abstract provides a clear and concise summary of the study, effectively outlining its objectives, methods, results, and conclusions. However, it would be better if additional details on how the system was evaluated, such as the number of qualitative interviews conducted are included in the methodology sub-section.

Response: Thank you, this has been clarified in the Methods section of the Abstract.

Comment 2: While the authors have provided a strong justification for AMR surveillance in introduction section, it would better to articulate the specific gaps in Kenya’s current AMU monitoring efforts, particularly in livestock farming, to emphasize the novelty and necessity of the study. Additionally, while the authors have briefly mentioned the risk of antimicrobial residues in animal products. The potential health consequences for consumers have not been discussed in enough detail. Moreover, while the role of ICT in AMU monitoring is introduced, there is limited discussion on previous digital interventions in similar settings or their potential limitations. Additionally, the transition from the problem statement towards the study’s objectives could be smoother to ensure a much natural flow from the broader AMR issue towards the specific rationale of piloting ADIS.

Response: Thank you for these suggestions. The absence of national AMU monitoring in Kenya has been clarified (L69-71)

A sentence about the negative health consequences of antimicrobial residues has been added with a new reference (L49-50).

The Kenya Animal Biosurveillance System application has been included briefly in the Introduction (L79-81) and in the Discussion (L328-332), as a relevant system for comparison (as it is used in Kenya, albeit by a different user group).

The transition to the study objectives has been rephrased to improve the flow of the text (L84-90).

Comment 3: In the results section, while the authors have presented a significant amount of numerical data but there are some inconsistencies in how certain values are reported (such as switching between percentages and absolute numbers), which could make interpretation challenging for readers. The sub-headings particularly those which summarize the farmer and agrovet perceptions, are quite descriptive but could benefit from a more structured analysis that groups findings into broader themes making it easier to identify key takeaways.

Response: We apologize for the inconsistencies in the descriptive data and have tried to include both numbers and percentages in most places, or in some sections only the total number and percentages, to clarify the results and facilitate the reading.

We have rephrased the headings and reorganized paragraphs 2.4. and 2.5. slightly to clarify and improve the flow of the text. The main themes that emerged in the focus group discussions are summarized in table 3. We hope that the edits now clarify that the text in the paragraphs summarize the agrovets’ perceptions and implementation of the tested system, while the table summarizes their general perceptions that emerged in the FGDs. As the farmer FGDs were more focused on their actual practices (in addition to their use of the ADIs system) than their general perceptions of medication and AMR, a similar summary could not be provided from these FGDs.

Comment 4: In the discussion section, while the authors have highlighted the critical role of farmers and agrovet operators in disease control and AMR prevention but they have not fully discussed the challenges of implementing syndromic surveillance such as potential misinterpretation of symptoms or reluctance among farmers to report cases accurately. The authors have also mentioned the economic implications for agrovet operators due to reduced antimicrobial sales but could have further discussed the potential policy or financial solutions to balance public health concerns with their livelihoods. The discussion section could also be more beneficial if a clearer linkage between the study’s findings and existing AMR policies in Kenya, as well as practical steps for integrating ADIS into national surveillance frameworks are discussed.

Response: Thank you for these suggestions, we have added some comments on the challenges with syndromic surveillance (L267-271) and managing the economic implications for agrovets (L323-326). We have also moved some sentences in the Discussion, to improve the flow of the text, and information about current and future policies as well as the next steps has been added (L335-339, L342-346).

Comment 5: The methodology section is well-structured. However, while the authors have described the study sites. But there is little discussion on why these specific counties were chosen beyond their proximity to Nairobi. The piloting of ADIS is also well explained but it is unclear whether any technical challenges were encountered such as issues with user compliance, data entry errors or system reliability. Additionally, the data collection process is mentioned but authors should also mention regarding any validation methods if applied to ensure accuracy in self-reported drug use and disease cases.

Response: Thank you, we have added a clarification/justification for the study site selection (L360-361). The technical challenges have been described in our previous paper, as clarified on L380-381). It was not possible to validate the accuracy of collected data. This will be the case also if the system were to be implemented as a national surveillance tool, as clarified in the discussion about syndromic surveillance (L268-269).

Comment 6In the conclusion section, the authors could also briefly mention the recommendations for future research or steps needed for scaling up the system such as refining the data collection process, increasing farmer participation, or integrating the system with existing surveillance programs.

Response: Thank you, this has been added to the Conclusion (L327-430)

Comment 7: I have also been noted that there are some grammatical issues in the manuscript. For example, authors should consider the subject-verb agreement error "Evidence from recent studies in Kenya show" and replace it with "Evidence from recent studies in Kenya shows". In the conclusion section "improvement of AMU" should be revised to "improvement in AMU". Similarly, "help policymakers to monitor" should be changed to "help policymakers monitor". Additionally, the phrase "and responsible AMU" is vague and would be clearer if it is modified as "and promote responsible AMU."

Comments on the Quality of English Language: The English could be improved to more clearly express the research.

Response: The entire text has been proof-read and the language has been improved for clarity and correctness.

Reviewer 2 Report

Comments and Suggestions for Authors

Experimental section: The same authors published a similar study on antibiotics in 2023. In that paper (Antibiotics 2023, 12(5), 905), the authors interviewed 100 farmers. However, they only interviewed 15 farmers in this paper. I suggest that the authors interview more farmers (at least 100).
2.2 Conclusion section
Figure 1: In the question about farmers' health reports, the authors also mentioned that "Ninety one (91) records were received from the farmers, 12 records had missing details and were excluded," which can easily confuse readers. What are the missing details? This section needs to be explained. If the data is incomplete, the sample data needs to be supplemented.
2.3 In lines 136 to 142, the author's explanation of farmers' visits to small farms is not smooth, and the author needs to revise this part. Especially lines 138 to 142, the explanation in this part does not seem to be substantive to what the author wants to express.
In lines 153 to 161, the author cited the commonly sold antibiotics drawings, and the data source and citation need to be stated.
3 Discussion
In this section, the author used a lot of text to discuss the results, but for example, in lines 231, 232, 233, and 234, the author analyzed and discussed the reasons why farmers visited agrovet outlets. These reasons can be simplified. Too much text will cause doubts and confusion to readers and peer scholars.
In addition, in terms of language and writing, I suggest that the author check the grammar and writing details of the whole article. I noticed that the order of some sentences in the article needs to be changed, and the language description method needs to be further improved.

Author Response

Comment 1: Experimental section: The same authors published a similar study on antibiotics in 2023. In that paper (Antibiotics 2023, 12(5), 905), the authors interviewed 100 farmers. However, they only interviewed 15 farmers in this paper. I suggest that the authors interview more farmers (at least 100).

Response: While we appreciate the validity of this suggestion, it is not possible to interview more farmers as we have included all that participated in the pilot study. Only 15 of the 100 farmers from our baseline study (Mutua et al, 2023) participated in the pilot testing of the system, while all veterinary pharmacies from the baseline took part in the pilot. As explained in Sternberg-Lewerin et al 2025, the numbers were seen as appropriate for the piloting of the system, as regular follow-up visits to provide technical support were foreseen. We have added a comment on the low number of farmers on L255-257.

Comment 2: 2.2 Conclusion section. Figure 1: In the question about farmers' health reports, the authors also mentioned that "Ninety one (91) records were received from the farmers, 12 records had missing details and were excluded," which can easily confuse readers. What are the missing details? This section needs to be explained. If the data is incomplete, the sample data needs to be supplemented.

Response: Thank you. As now clarified on L120-121, 12 records had missing details in the variable health problems (either only stating ‘new symptom’ or the name of a drug) and were excluded from the analyses including this variable

Comment 3: 2.3 In lines 136 to 142, the author's explanation of farmers' visits to small farms is not smooth, and the author needs to revise this part. Especially lines 138 to 142, the explanation in this part does not seem to be substantive to what the author wants to express.

Response: Thank you, sections 2.3-2.5 have been revised to improve clarity and flow of the text.

Comment 4: In lines 153 to 161, the author cited the commonly sold antibiotics drawings, and the data source and citation need to be stated.

Response: We are sorry, this must be a misunderstanding due to our mistakenly giving the wrong figure number here. There are no drawings, these are the antibiotics in the data, as presented in figure 4 (not 7).

Comment 5: 3 Discussion. In this section, the author used a lot of text to discuss the results, but for example, in lines 231, 232, 233, and 234, the author analyzed and discussed the reasons why farmers visited agrovet outlets. These reasons can be simplified. Too much text will cause doubts and confusion to readers and peer scholars. In addition, in terms of language and writing, I suggest that the author check the grammar and writing details of the whole article. I noticed that the order of some sentences in the article needs to be changed, and the language description method needs to be further improved

Response: Thank you, we have revised all of the text to improve clarity, and some sentences have been removed or shortened to facilitate the reading.

Round 2

Reviewer 1 Report

Comments and Suggestions for Authors

Authors have significantly improved the manuscript. I am now confident to recommend it for publication in its present form. 

Author Response

Thank you for this, and for your help

Reviewer 2 Report

Comments and Suggestions for Authors

The authors published a paper (Antibiotics 2025, 14(3), 285; https://doi.org/10.3390/antibiotics14030285) this month. In that paper, they also interviewed 15 farmers. Did the authors interview the same farmers in this manuscript? Did any data in this manuscript overlap with the data in their recently published paper?

Author Response

The referred manuscript overlap as regards the summary of the user perceptions of the system, but no detailed data are the same. Interviews at three months into the pilot study only appear in the referred paper. The farmer preparation study in the current manuscript, as well as data from the ADIS system and detailed description of the FGDs at the end of the pilot study only appear in the current manuscript. Thus, the two papers are complementary and not overlapping.

We have clarified (L342-343) that the previous paper includes general user perceptions. otherwise, as there are no overlapping or double reporting of data, we have chosen not to expand on this as it may confuse the reader.

Round 3

Reviewer 2 Report

Comments and Suggestions for Authors

I have no further comments.